# Analysis of the Dispersive and Distributive Mixing Effect of Screw Elements on the Co-Rotating Twin-Screw Extruder with Particle Tracking

**DOI:** 10.3390/polym16212952

**Published:** 2024-10-22

**Authors:** Jan Philipp Oldemeier, Volker Schöppner

**Affiliations:** Kunststofftechnik Paderborn, Paderborn University, 33098 Paderborn, Germany; volker.schoeppner@ktp.upb.de

**Keywords:** compounding, co-rotating twin-screw extruder, mineral fillers, dispersive mixing, distributive mixing, screw elements, 3D CFD flow simulation, particle tracking

## Abstract

Compounding is an important step in processing base polymers and is used to incorporate various additives into a polymer. For this purpose, different screw elements are used for dispersive and distributive mixing on a co-rotating twin-screw extruder. Optimising the screw configuration requires precise knowledge of the screw elements’ mixing properties, which have not been thoroughly investigated. This study analyses the mixing behaviour of individual screw elements regarding dispersive and distributive mixing using 3D CFD flow simulations with subsequent particle tracking. For distributive mixing, the particle distribution behind the screw elements in the XY plane is analysed and the mixing index MQ, which relates the standard deviation and the mean value of the triangular areas between the particles, is calculated. For dispersive mixing, the maximum shear stress on the particle path and the integral of the shear stress over the residence time of each individual particle are determined. The results show that screw element geometry and rotation speed have a significant influence on dispersive and distributive mixing. In addition, better dispersive mixing is achievable with highly viscous materials. These findings enable the optimisation of the mixing zone of a co-rotating twin-screw extruder for the efficient mixing of mineral fillers.

## 1. Introduction

During the compounding of polymers, base polymers are prepared in such a way that they fulfil the customer’s technical requirements. The properties of the polymer are specifically influenced by the addition of additives, fillers and colours [1,2,3].

Depending on the application, additives are selected, and the dosage is determined. Additives such as plasticisers, stabilisers, adhesion promoters, blowing agents and antioxidants are added to the polymer in small quantities of approx. 0.1 to 1 wt.%. Fillers, on the other hand, are added in larger quantities to produce more cost-effective plastic compounds. In addition to reducing costs, the properties of the compound can also be specifically influenced. Fillers are divided into the following two categories:Organic: wood, cellulose, carbon fibres, etc.Inorganic: talc, aluminium hydroxides, glass beads, etc.

The most important fillers in the plastics industry are grime, talc, chalk and barium sulphate.

Fillers can be categorised by the following physical properties: the aspect ratio, the particle size distribution and the specific surface area. The aspect ratio is a shape factor and indicates the ratio of the length to thickness of the filler particle. The different forms of the aspect ratio are shown in Table 1 [4,5].

The particle size distribution describes the distribution of the particle size with the specification of the particles present in a respective interval. This distribution can then be used to determine the average particle diameter, the top cut or the gradient of the particle size distribution curve [4,5].

The specific surface area [mm^2^/g] can be used to theoretically determine the number of adhesion points between the polymer chains and the aggregate. A large specific surface area represents a large number of adhesion points between the aggregate and the polymer and thus increases the stiffness, tensile strength and tear strength of the compound [4,5]. 

The incorporation of these fillers is usually realised with a co-rotating twin-screw extruder. This is mainly due to the flexibility of the twin-screw extruder thanks to the modular design of the barrel housing and the screws with the screw elements. The configuration of the barrels and screw elements can thus be perfectly adapted to the individual requirements of polymer compounding. The individual process zones of a twin-screw extruder are shown schematically in Figure 1. Firstly, the pellets are fed into the solids feed via a feed hopper of the twin screw extruder. The polymer is then melted in the plasticising zone. The filler is added to the filler feed zone and mixed with dispersive and distributive mixing in the mixing zone. Gases in the compound can be released into the environment in the degassing zone as well as being removed from the process with a vacuum. Finally, pressure is built up in the discharge zone to overcome subsequent dies [1,7,8,9].

Dispersive and distributive mixing is realised in the mixing zone with different screw elements, such as non-conveying kneading blocks, toothed mixing elements or screw mixing elements. The difference between the mixing types is that in dispersive mixing, the filler particles are crushed, whereas in distributive mixing, the filler particles are distributed. Screw elements such as the conveying and kneading elements of a twin-screw extruder are still based on classic self-cleaning Erdmenger profiles. Through further development of these screw elements, new non-self-cleaning profiles such as mixing, shearing and accumulation elements have been developed. Figure 2a–c shows some of these screw elements [1,11,12,13,14].

When configuring the screw in the mixing zone, the mixing performance with regard to dispersive and distributive mixing of the individual screw elements is particularly important and has therefore already been the topic of various research projects. As early as 1992, Manas-Zloczower and Yang used a 3D flow simulation to investigate the shear stress and the self-introduced flow parameter λ in the kneading disc region of kneading blocks in a co-rotating twin-screw extruder. They were able to show that the flow parameter λ increases with increasing stagger angle and that the rotational speed has an influence on the shear stress in the kneading disc region. They also showed that the shear stress and the parameter λ are the most important parameters for analysing mixing efficiency [15]. Building on this, Cheng and Manas-Zloczower analysed the mixing behaviour of two and three-flight kneading blocks in co-rotating twin-screw extruders in 1997 based on the shear stress that occurs and the components of the expansion flow. The two were able to show that the shear stress generated in the ZSK-53 with three-flight kneading blocks is greater than with two-flight kneading blocks on a ZSK-30 and therefore produces better flow conditions for dispersive mixing [16]. The evaluation of the mixing effect in plastic processing machines was taken up again in 2001 by Manas-Zloczower and Wang, and the mixing effect was analysed using particle tracking. In their study, it was shown that particle tracking can be used to analyse the course of each particle and the stress in the flow [17]. In 2007, Connelly and Kokini used particle tracking to analyse the mixing behaviour in single and twin-screw mixers and were able to show that twin-screw mixers are significantly better for mixing than single-screw mixers [18]. Zhang et al. investigated the mixing effect of kneading blocks in co-rotating twin-screw extruders in 2009 using numerical simulations and an experimental test. They analysed the residence time distribution and calculated mixing parameters such as the area expansion ratio, the instantaneous efficiency and the time-average efficiency. The results showed that the kneading block geometry can have a decisive influence on the residence time distribution and that kneading blocks with disc gaps, short disc lengths and larger stagger angles generally have good distributive mixing performance [19]. Nakayama et al. investigated the mixing effect of angled kneading blocks in 2010 [20]. The resulting agglomerate size of additives in co-rotating twin screw extruders was calculated in 2017 by Chen et al. using average deagglomerating energy. It was shown that there is an inherent connection between the screw combination and agglomerate dispersion, and a model for calculating the resulting agglomerate filler size was established [21]. In 2024, Frank developed a distributive and dispersive evaluation method for mixing processes on single-screw extruders based on particle tracking, a Delaunay triangulation and the calculation of the index according to Manas. Frank showed that Delaunay triangulation is suitable for assessing the distribution of dots on a surface and thus enables a statement about the distributive mixing effect [22]. In addition, other authors dealt with dispersive and distributive mixing on twin-screw extruders [23,24,25,26,27,28,29,30,31,32,33,34,35,36,37].

As the many scientific studies on mixing properties and the evaluation of these show, the topic is of great relevance. Yet, individual screw elements have not been fully analysed and compared with each other, so the design of a compounding process is still based on empirical values. The aim of this work is therefore to evaluate screw elements with regard to the dispersive and distributive mixing effect based on 3D CFD flow simulations and particle tracking. The two types of mixing are analysed separately. Distributive mixing is evaluated with a Delaunay triangulation and the mixing index MQ. For dispersive mixing, the maximum shear stress and the integral of the shear stress over the residence time for each individual particle are analysed.

## 2. Materials and Methods

The 3D CFD flow simulation was carried out with the following two polypropylene variants from SABIC Polymers (SABIC Polymers, Genk, Belgium):SABIC^®^ PP 505P, melt mass flow rate (MFR) 2 g/10 min at 230 °C and 2.16 kg;SABIC^®^ PP FPH50, MFR 50 g/10 min at 230 °C and 2.16 kg.

Figure 3 compares the viscosity curve over the shear rate of the two materials at 210 °C.

The following screw elements were selected for the investigations:Conveying element (SE), lead 24 mm, element length 24 mm;Kneading block (KB12), staggering angle 90°, number of kneading discs 5, element length 12 mm;Kneading block (KB24), staggering angle 90°, number of kneading discs 5, element length 24 mm;Kneading block (KB45), staggering angle 45°, number of kneading discs 5, element length 24 mm;Tooth mixing element (ZME), lead 8 mm, element length 16 mm.

The screw elements used are shown in Figure 4a–e.

The geometric parameters of the two following machines were used for the simulation: the ZSK25 (ZSK25, Coperion GmbH, Stuttgart, Germany), with a diameter of 25 mm, and the ZE28BP (ZE28BP, KraussMaffei Group GmbH, Parsdorf, Germany), with a diameter of 28 mm. On the process side, a melt temperature of 220 °C, a throughput of 20 kg/h for the ZSK25 and 22.4 kg/h for the ZE28BP and rotation speeds of 200, 300 and 400 rpm were set. The flow in the flow region was simulated for a total of 30 s. The 30 s were divided again into 3600 time steps so that one time step corresponded to 1/120 s. The points for the particle tracking were added from seed points with specified coordinates in the flow. For each seed point, 377 points were initialised per time step. Figure 5a shows the distribution of the seed points for the particles graphically. The two seed points were positioned in the centre of the channel height, offset by 90° to each other, positioned adjacent to the screw element, not covered by the screw element geometry and rotated at the same rotation speed around the axis of the screw elements. Because of the high number of time steps and the symmetry of the elements, only two seed points were selected. Figure 5b visualises the entire flow region with the inlet, the two seed points, the screw element and the outlet.

The flow simulation was thus carried out for 60 test points (2 machines × 2 materials × 3 rotation speeds × 5 screw elements) by the company IANUS Simulation GmbH (IANUS Simulation GmbH, Dortmund, Germany) and the results were made available as a CSV for analysis. The analysis was carried out using various Python 3.10 scripts, which were executed with the PyCharm programme (Community Edition 2023.1.3, JetBrains s.r.o., Prague, Czech Republic). For the evaluation of the distributive mixing quality, the X and Y position of the particles was determined at specified time steps behind the screw elements on an evaluation plane, and triangles were formed between the outer boundary (outer diameter), the two inner boundaries (inner diameter of the screws with a specified centre distance) and the particle points using Delaunay triangulation. The surface areas of the triangles were then calculated and the mixing index MQ was calculated. The mixing index was calculated as shown in Formula (1):(1)$$\mathrm{MQ}=\frac{s}{\bar{x}}\;$$
where “*s*” is the standard deviation of the surface area of the triangles formed and “$\bar{x}$” is the mean value of the triangle surface area. The lower the value for MQ, the more homogenous the surface area of the triangles and therefore also the distributive distribution of the particles. This evaluation was carried out for the following time steps: 1000, 1500, 2000, 2500, 3000, 3500 and 3600. The interval of the time steps was set to 500 time steps or approx. 4.16 s in order to ensure constant interval lengths. The values before 1000 were not considered so that the fluctuations at the start did not influence the results and a steady flow could therefore be assessed. The value at 3600 was also considered, as this was the value at the end of the simulation. For the analysis of the dispersive mixing, the maximum shear stress and the integral of the shear stress over the residence time were determined for each individual particle that completely flowed through the screw element. All 3600 time steps were taken into account, analysed in sequence and the maximum shear stress and the integral were updated. The values were then normalised and analysed in a histogram.

## 3. Results

### 3.1. Distributive Mixing

When analysing and interpreting the results for distributive mixing, the distribution of the particle points in the evaluation plane was graphically analysed in addition to the MQ index for the mixing quality. The calculated index MQ for the individual screw elements at 200 1/min, on the ZSK25 machine and with the material 505P are shown in Figure 6 as a function of the individual time steps.

The conveying element SE and the short kneading block KB12 lead to an MQ value of approx. 2.4 and, with a standard deviation of 0.11 for the kneading block and 0.15 for the conveying element, have approximately the same mixing effect as well as a standard deviation on the same scale. The MQ values of the three other screw elements (KB24, KB45 and ZME) are also close to each other and are all around 1.2. The standard deviation for the three elements varies between 0.06 (KB45), 0.03 (KB24) and 0.02 (ZME). At a rotation speed of 300 rpm, the temporal distribution of the MQ value is shown in Figure 7. The values for the KB12 and the SE are both still higher than the values for the three other screw elements, but the distance between the two ranges is halved from approx. 1.2 to approx. 0.6. The new MQ values for the KB12 and the SE are between 1.5 and 1.7. The standard deviation also becomes smaller and is 0.09 (KB12) and 0.14 (SE). The MQ values for KB24, KB45 and ZME are also slightly below the MQ values calculated at 200 rpm, but the distance between the three curves is increased. The ZME achieves the best MQ values of 0.92. The KB24 comes next with an MQ value of 1.02, and the KB45 is last with an MQ value of 1.16. The standard deviation for the KB24 and the ZME are lower and are now 0.027 (KB24) and 0.014 (ZME). Only the standard deviation for the KB45 is increased slightly to 0.07 compared with the value at 200 rpm.

Figure 8 below shows the MQ curves for 400 rpm. The MQ curves for the KB12 and the SE are still above the other curves at 1.34. However, the gap to the other curves has decreased again and is now only approx. 0.4. The curves for the ZME and the KB24 are at 0.9 (ZME) and 0.92 (KB24). The values of the KB45 are slightly above this at approx. 1.0.

The standard deviation is reduced for all curves and is now between 0.04 (KB12) and 0.008 (ZME). In addition, the MQ value is constant over time within a rotation speed range and thus, a constant mixing performance is generated with the screw elements. It is also clear that the standard deviation of the MQ values for all screw elements decreases as the screw speed increases. The curves of the mean values of MQ for the individual screw elements are shown in Figure 9. This graph clearly shows that increasing the rotation speed leads to a reduction in the MQ values. This relationship is independent of the screw geometry, but the magnitude of the influence of the rotation speed change on the MQ value depends on the screw element geometry. This becomes clear with the SE and the KB12, which are most strongly influenced by the change in screw rotation speed. The value for MQ decreases by 1.1 for both when increasing from 200 to 400 rpm. The mixing behaviour of the ZME, the KB24 and the KB45 are not so strongly influenced by the rotation speed. For these three screw elements, the value for MQ only decreases by approx. 0.3 with the same rotation speed change.

The different values for MQ are also recognisable in the graphical evaluation of the coordinates of the particles. Figure 10 shows the results of the Delaunay triangulation for the rotation speeds from 200 rpm (left) to 400 rpm (right) for the SE with the material 505P on the ZSK25. The green dots in this graph represent the particles, and the blue lines between the dots are the sides of the triangles created by the Delaunay triangulation. It is immediately noticeable that at 200 rpm, a lot of white areas are recognisable. Therefore the distribution of the green dots is much more uneven than, for example, at 400 rpm, as the proportion of white areas is greatly decreased there. In the distribution at 200 rpm, clear accumulations of particles can be seen in the centre left and top left, which are curved and directly next to the cylinder wall. However, there is a large area in the centre at the right at the same speed where there are no particles. At 300 rpm, the number of white areas decreases, but the particle accumulations remain. These particle accumulations are located at the top right and slightly below the centre on the right in the image of 300 rpm. As at 200 rpm, the particle accumulations are close to the cylinder wall. At 400 rpm, the particles are more evenly distributed, and the particle accumulations are smaller in size.

The graphics for the KB12 look similar to what is shown in Figure 10 and are illustrated in Figure 11. At 200 rpm, it is noticeable that one-half of the circle in the upper and lower circles contains particles, and the opposite side contains hardly any particles. This division of the circle areas is no longer present at 300 rpm, but particle accumulations near the cylinder wall can also be recognised here. These are very clearly recognisable in the lower left circle area. At 400 rpm, the particles are again distributed much more evenly, and the particle clusters exist both in the channel and near the cylinder wall but are no longer that large.

The results of the ZME, visualised in Figure 12, are in contrast to this. An even distribution of particles can already be seen at 200 rpm and, apart from a small accumulation of particles right below the centre, there are no accumulations of particles or areas with few particles. The accumulation of particles is no longer present at 300 rpm and the distribution of particles appears more even. The best distribution of particles is then achieved at 400 rpm.

The MQ value for the individual screw elements is only slightly influenced by the flow behaviour of the polymer. The extent of the effect of the material change on the MQ value also depends on the rotation speed and the extruder size. With the ZSK25 with the KB12, for example, there is no difference between the MQ value, but with the ZE28, there is a clear difference between the MQ values of the two materials, especially at 200 rpm. The curves of the MQ values as a function of rotational speed are shown for the different combinations of the extruder and material in Figure 13 for the KB12 and the ZME. The difference in MQ values between the two extruder sizes is the least at 400 rpm. With the ZME, this difference in the MQ values between the extruder sizes and the materials is different. For example, there is a roughly identical distance between the four different combinations at the beginning. However, this decreases with increasing rotation speed and almost completely disappears at 400 rpm.

Especially at low speeds, a slightly better MQ value can be achieved with the ZE28. The FPH50 can also achieve a slightly better MQ value compared with 505P. Graphically, this slight difference is difficult to see. As shown in Figure 14, only a few differences between the 505P and FPH50 materials are recognisable. The particle accumulation in the ZSK25 with FPH50 is smaller than in the ZSK25 with 505P, but it is still present. This difference is even smaller with the ZE28. Only a few white areas are no longer present when switching to FPH50. The influence of the extruder size, on the other hand, is clearly recognisable, as the particle accumulations are no longer present in the ZE28 in contrast to the ZSK25.

To summarise, the rotation speed has a major influence on the MQ value and therefore has a major influence on distributive mixing. The screw element geometry also has a major influence on the mixing effect and the MQ value. The toothed mixing element performs best, closely followed by the kneading block elements, and the conveying element is not suitable for distributive mixing. In contrast, the material and the extruder size only have a very small influence on the MQ value and therefore also on distributive mixing.

### 3.2. Dispersive Mixing

When evaluating dispersive mixing, the maximum shear stress of each particle on the path through the flow area and the integral of the shear stress on the path were analysed. These two values were used to assess the two mechanisms that can occur during dispersive mixing. These are breakage and erosion. For breaking the agglomerates, a force dependent on the material must be applied so that the agglomerates break up. Erosion, on the other hand, requires a lower force over a longer period of time.

#### 3.2.1. Maximum Shear Stress

Similar to distributive mixing, the screw element geometry has a major impact on the distribution of the maximum shear stress of the particles. The different influence on the frequency of the individual shear stress values is shown for the ZSK25 with the material FPH50 at 200 rpm in Figure 15a. The frequency distributions of the shear stress of all three kneading blocks have the first peak at approx. 18,000 Pa and then drop significantly. The conveying element only has its first peak in the frequency distribution at approx. 23,000 Pa and then also falls significantly. The frequency distribution of the ZME clearly stands out from the other frequency distributions, as it has a flat but wider plateau instead of a high peak. The plateau starts at approx. 28,000 Pa and extends to approx. 47,000 Pa. This means that the ZME generates a high shear stress for many particles. Compared with the other screw elements, the particles in the ZME are exposed to higher shear stresses. This becomes particularly clear when looking at the sum distribution of the shear stresses in Figure 15b. Where approx. 60% of the particles in the KB12, KB24 and SE elements have already reached the maximum shear stress, the sum distribution of the ZME is just beginning. The shear stresses are also more evenly distributed in the ZME, which can be recognised by the linear gradient of the sum distribution. For the elements KB12, KB24 and SE, on the other hand, the sum distribution rises sharply at the beginning and then flattens out continuously. For KB45, however, the course of the cumulative distribution is different. Up to approx. 30% of the sum distribution, the course of the sum distribution follows the course of the sum distributions of the KB12, SE and KB24. After that, the curve bends and rises again at higher shear stresses. From approx. 40%, the curve of the sum distribution is centred between the curves of the KB12, KB24 and SE on the one side and the ZME on the other.

The influence of the material on the distribution of the shear stress and the sum distribution are shown in Figure 16a,b. The frequency distribution of the shear stresses with the material FPH50 starts at approx. 25,000 Pa and ends at approx. 50,000 Pa. The material with a higher viscosity, 505P, experiences higher shear stresses. Here, the frequency distribution of the shear stresses starts at approx. 80,000 Pa and ends at approx. 130,000 Pa. This means that the frequency distribution of the shear stresses for the ZME at 200 1/min is twice as wide with the 505P material than with the FPH50 material. On the other hand, the relative frequency of the particles with the respective shear stress is accordingly only half as large. However, the geometric course of the frequency distribution of the shear stresses remains approximately the same, which can be seen in the linear courses of the sum distribution. Only the gradient is different because of the different widths of the frequency distribution of the shear stresses.

This shift in the shear stress curves when changing the material is recognisable for the ZSK25 for all combinations of rotation speed and screw element geometry. With the ZE28, on the other hand, the geometric course of the shear stress can also be influenced by the material change, especially at high speeds. These changes in the frequency distributions of the shear stresses are visualised in Figure 17a,b for a KB12 at 200 and 300 rpm. While the frequency distributions of the two materials are still comparable at 200 rpm, the frequency distribution of the shear stresses with the material FPH50 at 300 rpm differs significantly from the frequency distribution of the material 505P at the same speed.

The influence of rotation speed on the frequency distribution of shear stress is clearly recognisable for all screw elements, materials and extruder sizes. If the rotation speed is increased, the frequency distribution of the shear stress shifts to the right into larger shear stress ranges. This can be seen in Figure 18 for both the ZME (a) and the KB24 (b). However, increasing the rotation speed can also change the geometric course of the frequency distribution, as can be seen in Figure 18b. There, the peak still present at 200 rpm disappears almost completely at 300 rpm. This change in the geometric course of the frequency distribution of the shear stresses occurs in all kneading block variants used in combination with the material FPH50 when the rotation speed is increased from 200 rpm to 300 rpm.

As Figure 19a,b show, changing the extruder size from the ZSK25 to the ZE28 leads to greater shear stress generated in the flow area of the SE element. This correlation is present at all rotation speeds and materials.

#### 3.2.2. Integral of Shear Stress

In the following, the results for the integrals of the shear stress over the residence time are presented as a function of the screw elements, the rotation speed, the material and the extruder.

First, the effect of the screw element geometry on the frequency distribution of the shear stress integrals is analysed. As shown in Figure 20a,b, the lowest shear stress integrals are realised with the KB12. In the graph, approx. 60% of the shear stress integrals of the KB12 are less than 600,000 Pas. The KB45 also has its first peak at 500,000 and then again has a large amount of shear stress integrals that have a high value. The KB12, the KB24 and the SE generate the largest values for the shear stress integral. However, only a maximum of 10% of the shear stress integrals have values greater than 5,000,000 Pas. The ZME has a later peak here at approx. 1,600,000 Pas. The course of the frequency distribution of the shear stress integrals of the ZME is similar to a Gaussian curve. However, the maximum values for the shear stress integrals of the ZME are smaller than those of the KB24, KB45 and SE.

The influence of the material on the frequency distribution of the shear stress integrals is shown graphically in Figure 21a,b. In both graphs, it can be seen that larger shear stress integrals are achieved with the material 505P and that these are approximately directly linked to the range of the shear stress integrals of FPH50. The course of the frequency distribution of the integral of FPH50 has a large peak at approx. 1,600,000 Pa*s and then falls just as rapidly as it increased before the peak. In the case of the material 505P, the gradient at the peak of the frequency distribution of the shear stress integral and thereafter is not as steep as in the case of the material FPH50. This also becomes clear in the sum distribution in (b), as the gradient of FPH50 is very steep there.

The influence of rotation speed on the integrals of the shear stresses is similar to the influence of speed on the shear stresses. The integrals of the shear stress increase with increasing rotation speed. Thus, the first peak of the frequency distribution of the ZME at 200 rpm is still at 1,600,000 Pa*s, but at 300 rpm, it is already at 3,100,000 Pa*s and finally at 400 rpm at 4,700,000 Pa*s. In the case of KB24 in Figure 22b, the integral of the shear stress also increases with increasing rotation speed, but here again, the curve of the frequency distribution of the shear stress integrals is clearly influenced. The first peak at 200 rpm disappears completely, and the curve flattens out at 300 and 400 rpm. Figure 22a,b also clearly show that as the rotation speed increases, the curve of the frequency distribution of the shear stress integrals is stretched and thus, the slope to the peaks decreases.

The influence of the extruder size on the value of the shear stress integrals is illustrated in Figure 23. Figure 23a shows the relevant range of the frequency distribution of the shear stress integrals. The peaks of the shear stress integrals of the ZSK25 always lie before the peaks of the integrals of the ZE28 at the respective rotation speeds. As can be seen in Figure 23b, a change from the ZSK25 to the ZE28 also leads to higher final values of the shear stress integrals, and the number of integrals with high values increases.

### 3.3. Summary

The ZME proves to be the most suitable for distributive mixing, as both the MQ value, which relates the standard deviation and the mean value of the triangular areas between the particles, and the graphical analysis provide the best results for this screw element. The two kneading blocks KB24 and KB45 perform almost equally well and are slightly behind the ZME, where the KB24 performs slightly better than the KB45 for distributive mixing. The KB12 and SE perform significantly worse in terms of distributive mixing. In addition to the screw element geometry, the rotation speed also has a major influence on the distributive mixing behaviour of the individual screw elements. The lowest distributive mixing effect is achieved at low rotation speeds, while increasing the rotation speed improves the mixing effect. With the SE and KB12, the distributive mixing effect is significantly improved by increasing the speed. The material used, on the other hand, has only a small influence on the distributive mixing effect, depending on the rotation speed and extruder size. The situation is similar for the extruder size: at low rotation speeds, it still has a measurable influence on the mixing quality, but this influence is no longer measurable at high rotation speeds. The ZME is also very well suited for dispersive mixing, as many particles experience high shear stresses when flowing around the screw element. The KB45 generates slightly lower shear stresses but is still ahead of the other three screw elements. The lowest shear stresses are generated by the KB12. Similar to distributive mixing, increasing the rotation speed leads to higher shear stresses and thus supports dispersive mixing. Higher shear stresses are also generated for materials with a lower MFR value and the ZE28.

In terms of shear stress integrals, the ZME and SE perform equally well and have a wide range of high shear stress integral values. However, the ZME has lower maximum values than the SE. The KB24 and KB45 perform about the same, and the KB12 produces the lowest values for the shear stress integrals. Analogous to the shear stress, the integral of the shear stress increases with increasing rotation speed, a lower MFR value and a larger extruder.

The disperse mixing behaviour of the individual screw elements can be analysed for the given process parameters using the curves of the maximum shear stress and the shear stress integral. It should be noted that higher maximum shear stresses support the break-up of agglomerates, whereas higher values for the shear stress integrals play a more important role in erosion. If the dispersive mixing quality of a compound is insufficient, these distributions can be used to adjust the mixing process and the resulting mixing quality either by adjusting the process parameters or by changing the screw configuration so that the filler has a more uniform particle size distribution and smaller particles after the mixing process.

## 4. Discussion

When comparing the results of this study with those from other publications, it becomes clear that particle tracking delivers comparable results. Manas (1992) already reported that higher rotational speeds lead to greater shear stresses and that the offset angle of kneading blocks only has a minor influence on the shear stresses. The simulations with particle tracking showed that higher rotation speeds lead to greater shear stresses in the individual particles. It was also found that the offset angle has a small measurable influence on the frequency distribution of the shear stresses in this study. In several studies, Fard and Anderson, Zhang et al. and Kohlgrüber reported that a kneading block with a 90° offset angle achieves a better mixture than one with a 45° offset angle. This was confirmed by particle tracking and Delaunay triangulation. The simulations showed that the KB24 achieves better distributive mixing than the KB45. For example, at 400 rpm on the ZSK25, the MQ value of the KB24 with the material 505P is 0.92 with a standard deviation of 0.011, while the KB45 only achieves an MQ value of 0.99 with a standard deviation of 0.029 under the same process parameters. However, the large difference between the two kneading blocks regarding dispersive mixing described by Kohlgrüber could not be fully reproduced. Although there was a difference in the shear stress distribution depending on the offset angle of the kneading block discs, this difference was less noticeable in the frequency distributions of the shear stress integrals.

Particle tracking is suitable for evaluating the mixing properties of individual screw elements for the twin-screw extruder, as the position in the flow area and the shear stresses on the particle path are known for each individual particle. While distributive mixing can be well represented by Delaunay triangulation, it should be noted that the MQ index does not consider either the screw element length or the number of points, which limits the direct comparison and must be considered to avoid misinterpretations. For dispersive mixing, particle tracking offers the possibility of determining the maximum shear stress and the shear stress integral over the residence time of each individual particle. These two parameters make it possible to evaluate the dispersive mixing effect of the individual screw elements and facilitate the comparison between different screw elements. However, the simulation results have not yet been confirmed experimentally. To validate the results of the simulations, experimental tests are currently being carried out. This involves the use of a special sampling plate that allows samples to be taken from the process across to the extrusion direction. Two of these plates will be used to take and analyse samples in front of and behind the investigated screw element.

## 5. Conclusions

In this article, the dispersive and distributive mixing properties of five different screw elements (SE, KB12, KB24, KB45 and ZME) of a co-rotating twin screw extruder were analysed at the University of Paderborn (Kunststofftechnik Paderborn, Paderborn, Germany) using 3D CFD flow simulations and particle tracking carried out by the company IANUS Simulation GmbH (IANUS Simulation GmbH, Dortmund, Germany). In particular, the influence of the screw element geometry, the rotation speed, the material and the extruder on the distribution of the particles in a cross-sectional plane for the distributive mixing behaviour and the maximum values for the shear stresses and the shear stress integrals along the particle paths travelled for the dispersive mixing behaviour were analysed. The results show that the screw element geometry and the rotation speed have a significant influence on the distributive mixing behaviour. The ZME proved to be particularly effective for distributive mixing in this study, with an increase in rotation speed further improving the mixing behaviour. Increasing the rotation speed from 200 rpm to 400 rpm improves the already very good value for the MQ index of the ZME from around 1.17 at 200 rpm to around 0.9 at 400 rpm. In comparison, the values for the KB12 at 200 rpm are around 2.38 and are improved to a value of 1.34 by increasing the speed to 400 rpm. Thus, doubling the rotation speed of the KB12 even leads to an improvement in the MQ value by 1. The influence of the material and extruder, on the other hand, is rather small for the distributive mixing behaviour. The dispersive mixing behaviour is also strongly influenced by the geometry of the screw elements and the rotation speed. Here, the ZME again proves to be the most suitable, and a higher rotation speed leads to higher values in the maximum shear stress and the shear stress integrals. For example, increasing the rotation speed of the ZME by 100 rpm on the ZSK25 with the material FPH50 leads to an increase in the mean value of the maximum shear stress by approx. 8300 Pa while the standard deviation remains the same. The mean value of the maximum shear stress of the ZME is therefore increased from 37,900 Pa at 200 rpm to approx. 54,400 Pa at 400 rpm. In comparison, with the KB24 with the same material on the same machine, the mean value of the maximum shear stress increases by approximately 10,300 Pa per 100 rpm and thus rises from 27,900 Pa at 200 rpm up to 48,400 Pa at 400 rpm. However, the standard deviation for the KB24 increases by approximately 1300 Pa. For the integral of the shear stress over the residence time, the mean value is increased by approximately 1,859,000 Pa for the ZME with a rotation speed increase of 100 rpm. In comparison, the mean value for the shear stress integral increases by 2,690,700 Pas for the KB24 with the identical rotation speed increase, but the standard deviation also increases by a factor of 2.7 as a result. With the ZME, this increase in the standard deviation is smaller, but still approx. 2.4 times the initial standard deviation. However, the standard deviation for the ZME is significantly lower than for the KB24. At 400 rpm, the ZME achieves a value for the standard deviation that the KB24 already has at 200 rpm. In addition, it became clear that materials with low MFR values and a change from the ZSK25 to the ZE28 can contribute to an increase in the maximum shear stress and the shear stress integrals, which has a positive effect on dispersive mixing. In the future, these methods will be used to evaluate the dispersive and distributive mixing properties of screw elements. These evaluated screw elements will then be used to design mixing processes or optimise existing mixing processes for the fillers used.

## Figures and Tables

**Figure 1 polymers-16-02952-f001:**
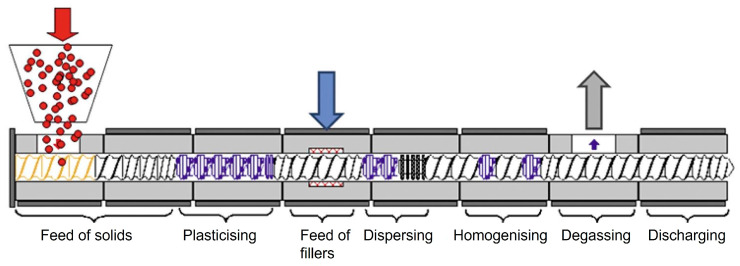
Processing zones of a twin-screw extruder with a polymer (red), a filler (blue) and an emerging gas (gray) [10].

**Figure 2 polymers-16-02952-f002:**
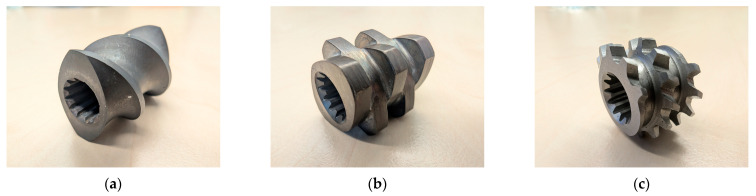
(**a**) Normal conveying element, (**b**) kneading block element with 5 kneading discs and a staggering angle of 90° between the kneading discs, and (**c**) a toothed mixing element.

**Figure 3 polymers-16-02952-f003:**
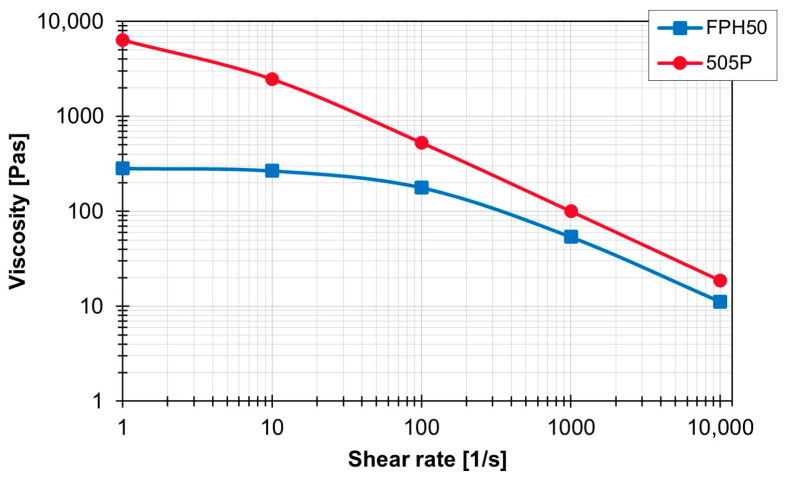
Viscosity of the materials used as a function of the shear rate at 210 °C.

**Figure 4 polymers-16-02952-f004:**

(**a**) Visualisations of the screw elements used: (**a**) SE, (**b**) KB12, (**c**) KB24, (**d**) KB45, and (**e**) ZME.

**Figure 5 polymers-16-02952-f005:**
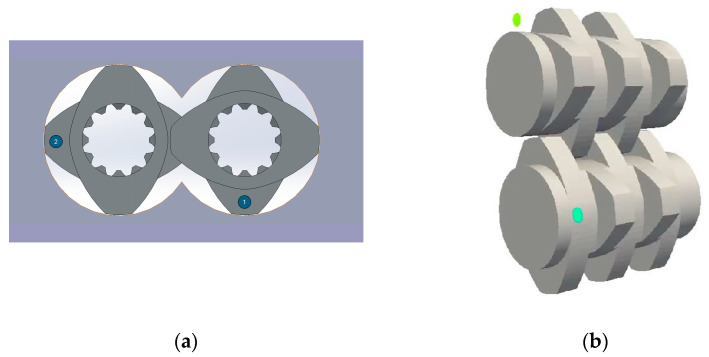
(**a**) Position of the two seed points (1, 2). (**b**) Overview of the flow area with an inlet, the screw element and an outlet.

**Figure 6 polymers-16-02952-f006:**
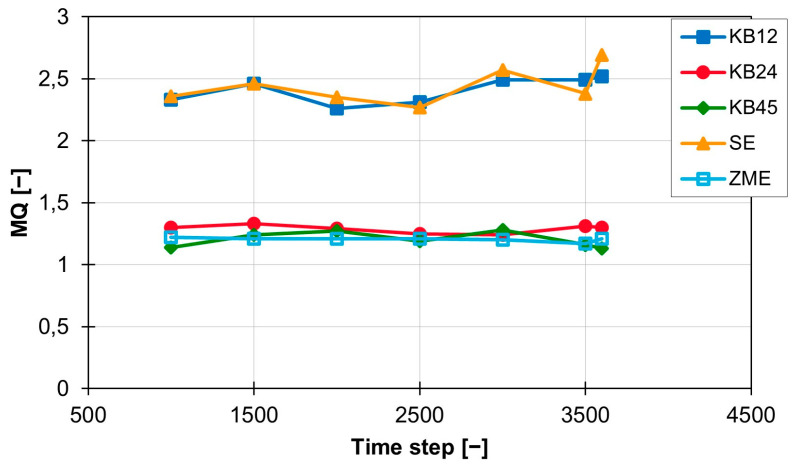
Time curve of the MQ index for the individual screw elements at 200 rpm with the material 505P on the ZSK25.

**Figure 7 polymers-16-02952-f007:**
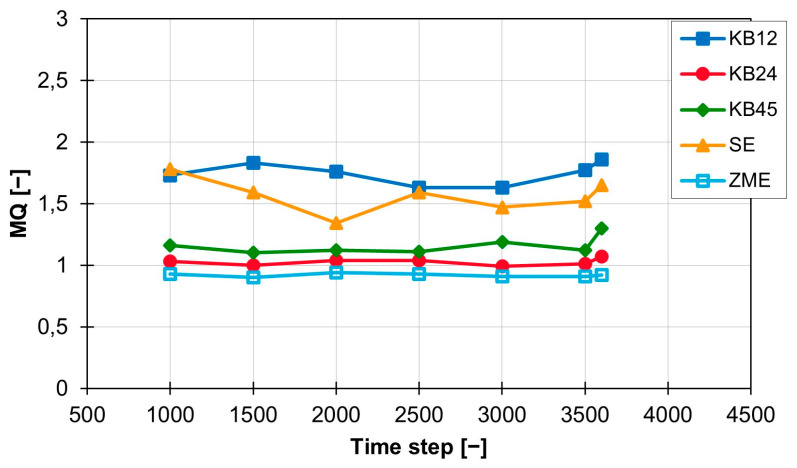
Time curve of the MQ index for the individual screw elements at 300 rpm with the material 505P on the ZSK25.

**Figure 8 polymers-16-02952-f008:**
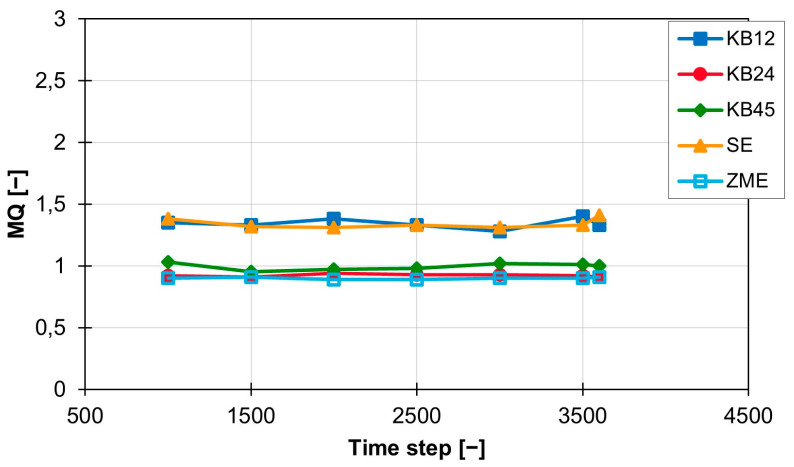
Time curve of the MQ index for the individual screw elements at 400 rpm with the material 505P on the ZSK25.

**Figure 9 polymers-16-02952-f009:**
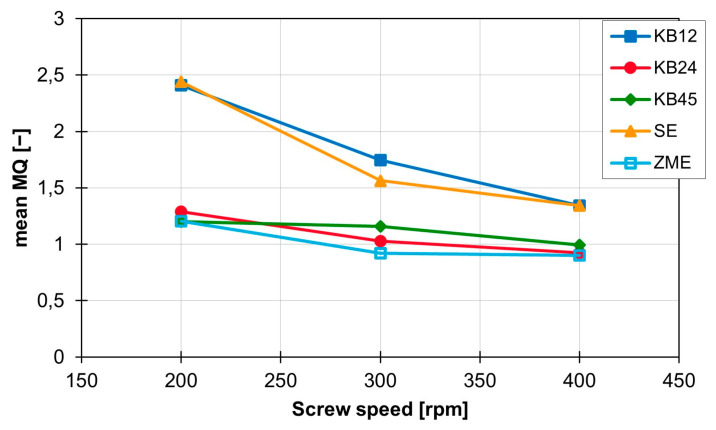
Influence of the screw speed on the characteristic mean MQ value for the individual screw elements on the ZSK25 with the material 505P.

**Figure 10 polymers-16-02952-f010:**
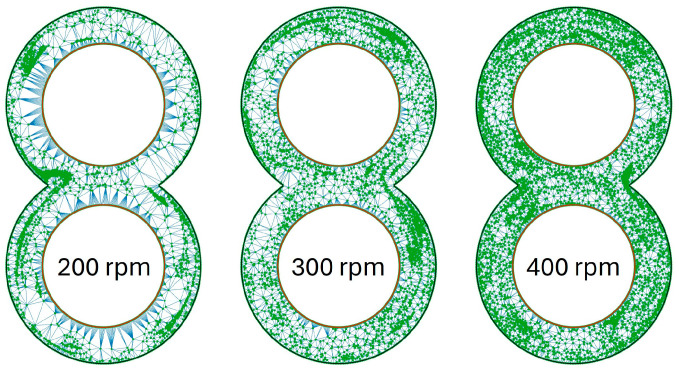
Distribution of particles in the evaluation plane for the SE at time step 3500 with the material 505P on the ZSK25 as a function of rotation speed.

**Figure 11 polymers-16-02952-f011:**
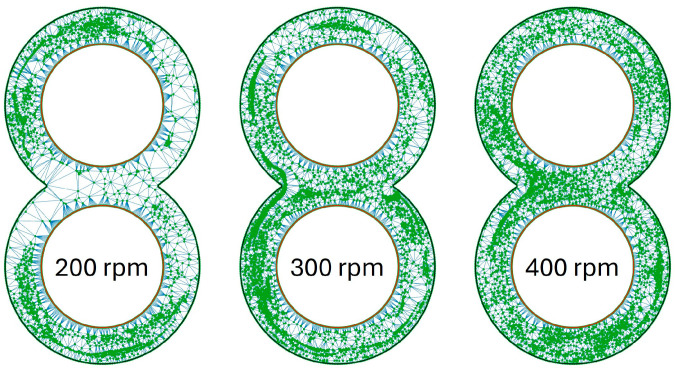
Distribution of particles in the evaluation plane for the KB12 at time step 3500 as a function of rotation speed.

**Figure 12 polymers-16-02952-f012:**
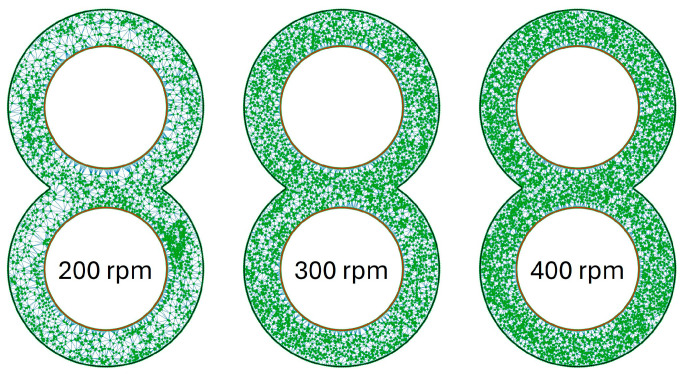
Distribution of particles in the evaluation plane for the ZME at time step 3500 as a function of rotation speed.

**Figure 13 polymers-16-02952-f013:**
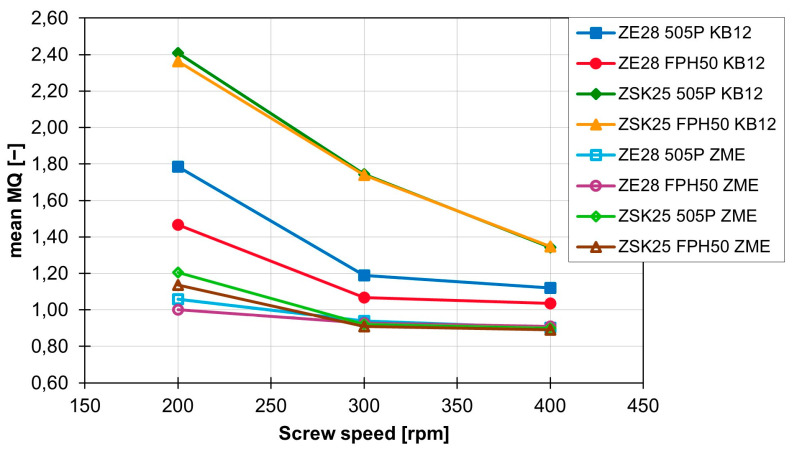
Development of the MQ values for the ZME and the KB12 for the different extruder sizes and materials as a function of rotation speed.

**Figure 14 polymers-16-02952-f014:**
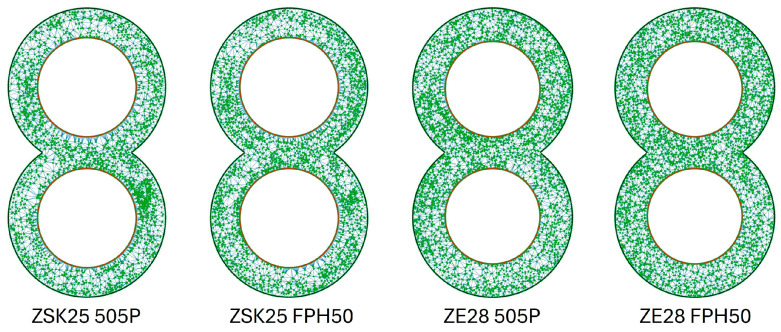
Graphical representation of the point coordinates for the ZME at 200 rpm as a function of the extruder size and the material.

**Figure 15 polymers-16-02952-f015:**
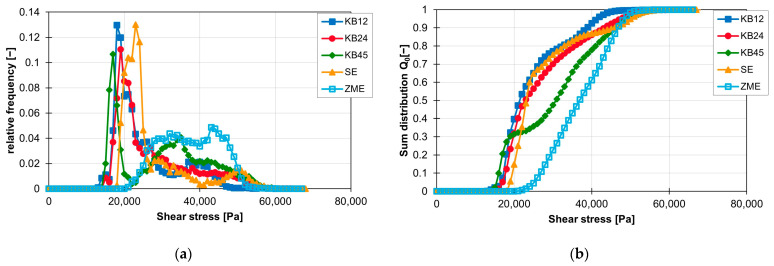
Graph of the relative frequency of shear stress (**a**) and graph of the sum distribution of shear stress (**b**) for the individual screw elements on the ZSK25 with the material FPH50 at 200 rpm.

**Figure 16 polymers-16-02952-f016:**
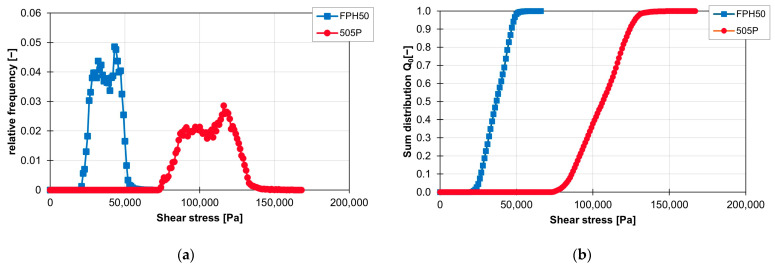
Graph of the relative frequency of the shear stress (**a**) and graph of the sum distribution of the shear stress (**b**) for the ZME on the ZSK25 at 200 rpm with the two materials FPH50 and 505P.

**Figure 17 polymers-16-02952-f017:**
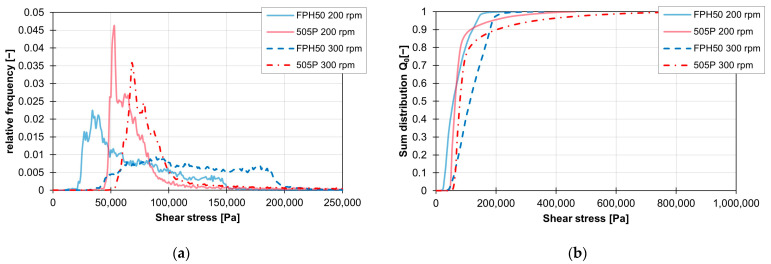
Graph of the relative frequency of the shear stress (**a**) and graph of the sum distribution of the shear stress (**b**) for the KB12 on the ZE28 at 200 and 300 rpm with the two materials FPH510 and 505P.

**Figure 18 polymers-16-02952-f018:**
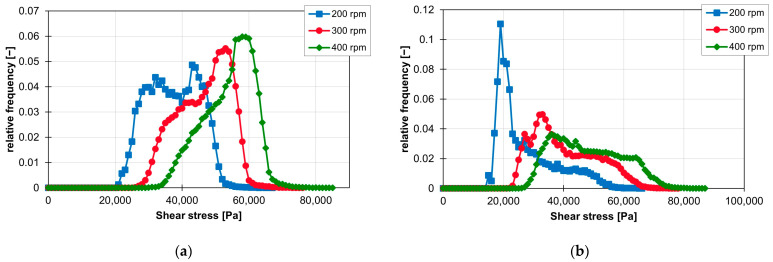
Influence of rotation speed on the resulting shear stresses for the ZME (**a**) and the KB24 (**b**) on the ZSK25 with the material FPH50.

**Figure 19 polymers-16-02952-f019:**
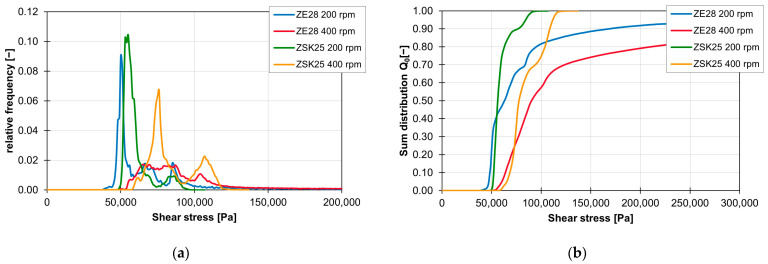
Influence of the extruder size on the shear stress curves of the screw conveying element at 200 to 400 rpm and the material 505P, shown by the relative frequency (**a**) and the sum distribution (**b**).

**Figure 20 polymers-16-02952-f020:**
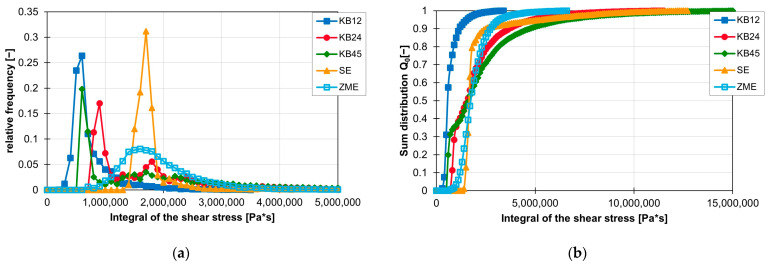
Influence of the screw element geometry on the frequency distribution of the shear stress integral on the ZSK25 with the material FPH50 at 200 rpm, shown by the relative frequency (**a**) and the sum distribution (**b**).

**Figure 21 polymers-16-02952-f021:**
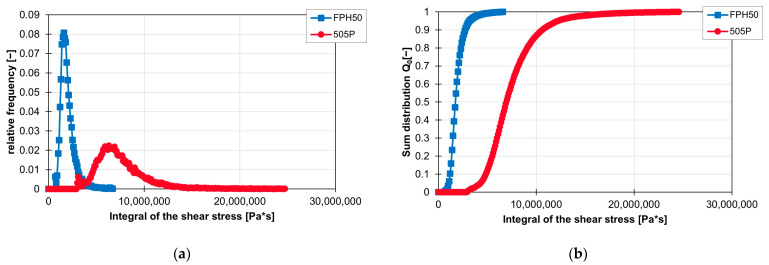
Graph of the relative frequency of the integral of the shear stress (**a**) and graph of the cumulative distribution of the integral of the shear stress (**b**) for the ZME on the ZSK25 at 200 rpm with the two materials FPH50 and 505P.

**Figure 22 polymers-16-02952-f022:**
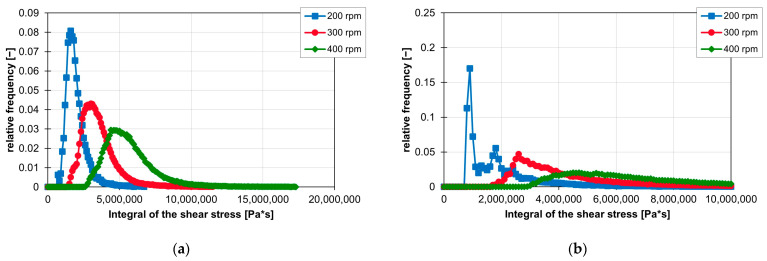
Influence of rotation speed on the resulting integrals of the shear stresses for the ZME (**a**) and the KB24 (**b**) on the ZSK25 with the material FPH50.

**Figure 23 polymers-16-02952-f023:**
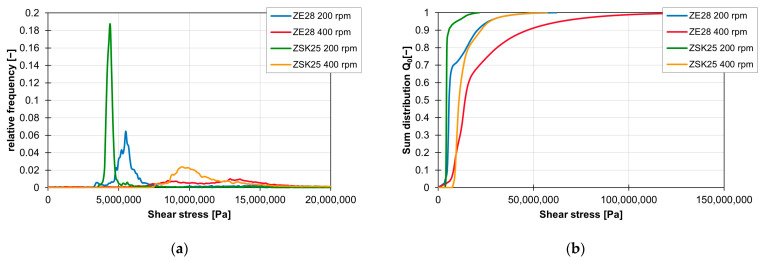
Influence of the extruder size on the shear stress integrals of the screw conveying element at 200 to 400 rpm and the material 505P, shown by the relative frequency (**a**) and the sum distribution (**b**).

**Table 1 polymers-16-02952-t001:** Classification of fillers based on the aspect ratio [6].

Shape	Aspect Ratio	Examples
Sphere	1	Glass spheres
Cube	1	Calcium carbonate
Cuboid	2–4	Silicates
Plates	5–10 (100)	Talcum
Fibres	<10	Glass fibres

## Data Availability

The data that support the findings of this study are available from the corresponding author upon reasonable request.

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
