# Peer review of "Analysis of the Dispersive and Distributive Mixing Effect of Screw Elements on the Co-Rotating Twin-Screw Extruder with Particle Tracking"

_polymers, 2024, doi:10.3390/polym16212952_

Round 1
Reviewer 1 Report
Comments and Suggestions for Authors
The comments are provided in the attached file.

Minor editing of references must be performed as detailed in the comments to the authors.
Reviewer 2 Report
Comments and Suggestions for Authors
In this research, authors analyzed and evaluates the mixing behavior of individual screw elements regarding dispersive and distributive mixing. The results gathered in this work made it possible to optimize the mixing zone of a co-rotating twin-screw extruder for the dispersive and distributive mixing of mineral fillers. Sufficient characterizations and measurements were performed; however, detailed scientific and theoretical analysis of the experimental results were not enough.
This manuscript will be considered to be published in Polymers after minor revision. Some doubts and comments are listed below.
1. In the Abstract, “The results showed that the screw element geometry and the rotation speed have a significant influence on the dispersive and distributive mixing behaviour of the screw elements….”. The authors should conclude some qualitative and quantitative results to explain how the screw element geometry and rotation speed influence the dispersive.
2. For the influence of 400rpm on distributions of particles in figure 10, please explain why doesn't the accumulation of particles occur near the cylinder wall?
3. For figure 16, it is stated that from the perspective of material properties, 505P has higher viscosity and shear stress, and its shear stress distribution is also shifted to the left. However, in Figure 17a, why is the distribution of shear stress for both materials not obvious at a rotational speed of 300rpm?
4. Most of the references were 8-10 years ago, please update.
Comments on the Quality of English LanguageMinor editing of English language required.
Round 2
Reviewer 1 Report
Comments and Suggestions for Authors
Authors have addressed all the comments raised in the 1st round of review.
Reviewer 2 Report
Comments and Suggestions for Authors
In my opinion, the revised manuscript can be considered to be published.
Comments on the Quality of English LanguageMinor editing of English language required.